



# Rescue, Integration, and Analytical Application of historical data from eight pioneering geomagnetic observatories in China

Suqin Zhang[1], Changhua Fu[1], Jianjun Wang[2], Chuanhua Chen[3], Guohao Zhu[4], Qian Zhao[5], Jun Chen[4], Shaopeng He[6], Bin Wang[6], Pengkun Guo[6], Na Deng[7], Jinghui Lu[8], Hongchi Yu[9]

[1]Institute of Geophysics, China Earthquake Administration, Beijing, 100081, China
[2]Earthquake Administration of Gansu Province, Lanzhou, 730000, China
[3]Earthquake Administration of Shandong Province, Jinan, 250014, China
[4]Shanghai Earthquake Agency, Shanghai, 200062, China
[5]Earthquake Administration of Liaoning Province, Shenyang, 110034, China
[6]Hebei Earthquake Agency, Hebei Province, Shijiazhuang, 050022, China
[7]Earthquake Administration of Hubei Province, Wuhan, 430071, China
[8]Earthquake Administration of Guangdong Province, Guangzhou, 510070, China
[9]Earthquake Administration of Jilin Province, Guangzhou, 130117, China

*Correspondence to*: Changhua Fu (fuchanghua2004@163.com)

**Abstract.** Decades to centuries of continuous geomagnetic observation data have extensive scientific research and practical application value, especially in revealing the long-term variation rules of the geomagnetic field, which is irreplaceable. During the International Geophysical Year (1957-1958), China established geomagnetic observatories in Beijing, Lhasa, Lanzhou, Wuhan, Guangzhou, Changchun, and Urumqi, forming the initial structure of China's geomagnetic observation 20 network together with the Shanghai Observatory. These observatories have continuously observed despite facing many challenges since their establishment, accumulating a large amount of valuable observational data, making significant contributions to the progress of geomagnetic scientific research and development. However, the scattered storage state of these historical data and the potential risk of damage pose a threat to the integrity and reliability of the data. This study conducted a rescue integration of the historical observational data from eight pioneering geomagnetic observatories in China, 25 significantly improving data quality and facilitating long-term preservation and use of the data. This article introduces the basic conditions of eight observatories, including their locations, changes in location, observation environments, magnetic rooms, the magnetism of building materials, the layout of building facilities, measuring instruments, etc. These are the main prerequisites and foundations for ensuring the quality of observation data. Then, it introduces the integration and processing of historical data, including data collection, digitization, unification of formats, anomaly detection, and 30 processing. Subsequently, the processed data were validated, including assessments of daily variations accuracy and long-term stability. The results show that the quality of the integrated historical data has been significantly improved. These datasets are of great value for improving historical geomagnetic field models, studying variable fields, main geomagnetic fields, and their long-term variations. Finally, we applied the data to the analysis and research of Sq and geomagnetic jerks, exploring the spatiotemporal variation characteristics of Sq and jerk in the China. Sq is mainly a daytime phenomenon, and its 35 variation pattern in the middle and low latitude regions is mainly characterized by its dependence on latitude and local time. The geomagnetic jerk phenomenon exhibits significant regional differences and asynchronous occurrence times of jerks. Jerk events in 1969, 1979, 1991, 2003, and 2019 were observed at all observatories and had distinct jerk variation characteristics. Other jerks were only observed at some observatories or individual observatories. The maximum time difference for the occurrence of the same jerk event at different observatories was 2 years. This study aims to provide these precious datasets to 40 the scientific community and the public so that these data can be integrated with data from other sources, thereby further exploring the spatiotemporal evolution and physical mechanisms of the geomagnetic field. The historical datasets of the eight geomagnetic observatories that have been integrated and quality controlled are available at https://doi.org/10.5281/zenodo.14560950 (Zhang et al., 2024b).



## 1 Introduction

Geomagnetism is an observational science. Geomagnetic observation data contains rich information about the near-Earth space. By studying this information, we can not only gain a deep understanding of the physical properties and dynamics within the Earth, but also investigate the structure of the magnetosphere and ionosphere, as well as the impact of solar activity on the human living environment, which holds significant importance. The application scope of geomagnetic observation data is extensive, covering fields such as aerospace, communication, mineral resource exploration, underground pipeline networks, power systems, oil drilling, space weather forecasting, and earthquake prediction (Reay et al., 2005; Marshall et al., 2011; Bolduc et al., 2002; Boteler et al., 1998; Liu et al., 2008, 2016; Liu et al., 2009), attracting widespread attention from scientists around the world. In particular, constructing historical geomagnetic models and researching the evolution of the Earth's magnetic field and its related dynamic processes largely depend on precise measurement data of long-term changes in the geomagnetic field provided by geomagnetic observatories (Clarke et al., 2009; Soloviev, et al., 2018; Gillet et al., 2013). Therefore, establishing and maintaining permanent geomagnetic observatories with a certain density and uniform distribution is crucial.

In 1834, German scientist Gauss established the world's first geomagnetic observatory to measure the three-component absolute values of the geomagnetic field—the Göttingen Geomagnetic Observatory. The establishment of the Göttingen Geomagnetic Observatory marked the beginning of systematic observation and research of geomagnetic field changes. To gain a more comprehensive understanding and mastery of the change patterns of the geomagnetic field, scientists realized the necessity of establishing more observatories worldwide. Currently, over 130 geomagnetic observatories globally have joined INTERMAGNET worldwide, providing continuous, quasi-real-time online data (http://www.intermagnet.org). These observatories are distributed around the world, forming a vast network for geomagnetic field observations.

In China, the first geomagnetic observatory was established in Beijing by Russian scientists in 1870 but ceased operations in 1882. In 1874, French missionaries established a geomagnetic observatory in Xujiahui, Shanghai, which was relocated to Lujiazui in 1908 and then to Sheshan in 1932, where it has continued to operate until now, becoming one of the longest-running geomagnetic observatories globally with nearly 150 years of continuous observation records. Additionally, the United Kingdom and Germany set up geomagnetic observatories in Hong Kong and Qingdao respectively. From the 1930s to the 1940s, Chinese scholars established observation observatories in places such as Zijin Mountain in Nanjing, Liangfeng in Guangxi, Guilin, and Beibei in Chongqing, but these observatories ceased operations before 1944.

During the International Geophysical Year, geomagnetic observatories were established in Beijing, Lhasa, Lanzhou, Wuhan, Guangzhou, Changchun, and Urumqi by China, together with the Shanghai Observatory, forming the eight pioneering geomagnetic observatories in China, collectively known as the "Old Eight observatories." The layout of these observatories is roughly cross-shaped (Figure 1), forming the basic framework of geomagnetic observation network of China(Cheng,

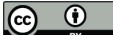



1995). They undertake the critical task of monitoring the spatiotemporal changes of the geomagnetic field in China, laying the foundation for the construction and development of China's geomagnetic network, and making significant contributions to the geomagnetic scientific research and development in China. In 1984, the "Old Eight observatories" were awarded the"International Centennial Geomagnetic Observation Commemorative Medal" by the International Association of

Geomagnetism and Aeronomy (IAGA) in recognition of their outstanding contributions in the global field of geophysical observation. As the "Old Eight observatories" with a long history of observations, they have accumulated a large amount of valuable data, becoming an important foundation for research in geomagnetism. However, historical materials are scattered, with a large portion existing in paper form, which is extremely inconvenient to use. Moreover, due to the long storage period, complex and harsh preservation conditions, these precious materials are facing severe risks of damage. Over time, these

materials may be lost or damaged due to natural wear and tear, thus affecting their application in scientific research. Therefore, the rescue work of historical materials has always been a focus of attention in the global scientific community (Jonkers et al., 2003; Chulliat and Telali., 2007; Dawson et al., 2009; Dong et al., 2009; Korte et al., 2009; Reay et al., 2013; Morozova et al., 2014, 2021; Zhao et al., 2017; Sergeyeva et al., 2021; Zhang et al., 2022a).

Thanks to the advancement of the national special project for the rescue of historical data in the field of earthquakes by the

China Earthquake Administration,we have carried out the rescue, organization, and analysis of these historical data First, we collected observational data that was stored in various locations and in multiple media formats. Then, we digitized the paper materials using digital photography and manual entry, and stored data from different periods and formats in a unified database for management. Subsequently, we conducted quality checks and processing on the data in the database, and validated the processed data. Finally, we applied and analyzed these data. Systematically organizing and summarizing these

data is of great significance for understanding the long-term evolution patterns of the geomagnetic field and and it also provides valuable data support for future related research.

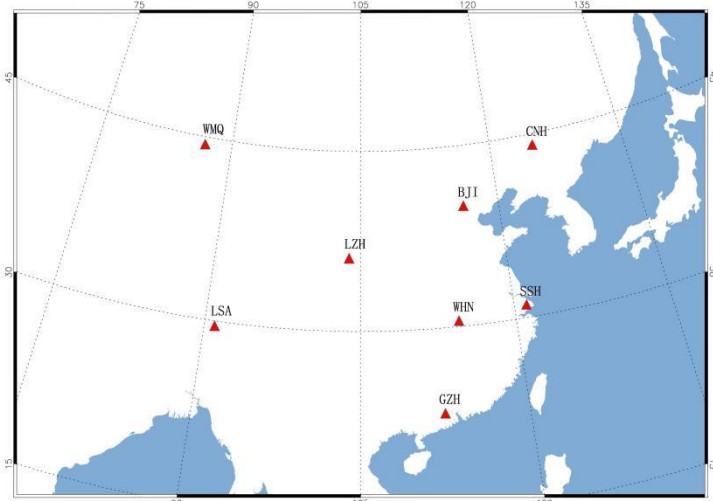

**Figure 1: Distribution map of the eight pioneering geomagnetic observatories in China**



## 2 Data Acquisition

Ensuring data quality is crucial for the reliability of research outcomes and the effectiveness of their application. The quality of geomagnetic data is influenced by a variety of factors, including the regional geomagnetic background conditions of the observation site (such as the foundation, topography, etc.), Magnetic field gradient of the observation site, the magnetic properties of the building materials used, the layout of the magnetic room, the insulation of the observation room, the configuration and updating of observation instruments, measurement arrangements, and data processing, among others

(Jankowski and Sucksdorff 1996; CEA, 2004; Kudin, 2021; Linthe et al. 2013; Zhang et al. 2024a). This study will provide a detailed introduction to the geomagnetic observation conditions at various observatories, including observatory site selection and historical changes, magnetic environment of the observatories, building materials, magnetic room layout, insulation conditions of the observation room, and the configuration of observation instruments.

### 2.1 Basic Information of observatories

#### 2.1.1 Changchun observatory (IAGA code CNH)

The Changchun observatory, started construction in 1951 and is located in the Nanling area (43.8°N,125.3°E) in the southern suburbs of Changchun City, Jilin Province. The observatory was completed in 1954 and underwent instrument installation and trial observations from 1954 to 1956. Formal observation work began in 1957. The geology foundation of the observatory and surrounding area is composed of Quaternary alluvial layers, approximately 30 meters thick, consisting of loess and sub-

clay.The magnetic field around the observatory is homogeneous, with a horizontal gradient less than 0.5 nT per meter. The absolute measurement house and variation recording room are integrated, with the absolute measurement house being a ground-level building and the recording room a semi-underground structure. It is built using weak magnetic materials such as limestone, wood, and copper.The range of the annual temperature in the recording room is 16℃, and the range of the daily temperature is 0.5℃.

Due to the rapid urbanization of Changchun City, the geomagnetic observation have inevitably been disturbed. Therefore, at the end of 1978, the Changchun observatory was relocated to Helong Town, Nong'an County, northwest of Changchun City (latitude 44.0°N, longitude 125.2°E). The geology foundation of the observatory and surrounding area is Cretaceous strata, consisting of mudstone, shale, and fine sandstone. The magnetic field around the observatory is homogeneous, with a horizontal gradient less than 1 nT per meter. The absolute measurement house is a ground-level building, and the recording

room is a semi-underground building. Both are constructed using weak-magnetic materials such as limestone, white bricks, pearl powder, copper, and wood. The annual temperature difference in the recording room is about 24℃,and the daily temperature difference is about 0.5℃.

Unfortunately, the construction of rail transit has once again caused interference to the geomagnetic observation. Consequently, the Changchun observatory was relocated to Sangan Township, Nong'an County (latitude 44.1°, longitude

125.0°) in 2007, and began normal observation on June 1st of the same year. The observatory is built on a base of Cretaceous



Songhua River Group thick sedimentary strata, consisting of non-magnetic and weakly magnetic mudstones, siltstones, and sandstones, with a horizontal gradient of the ground magnetic field less than 1nT per meter. Both the absolute measurement house and the relative recording room are constructed with weak magnetic materials. The measurement house is a ground-level building, while the recording room is a fully underground structure, with loess covering providing natural insulation. The annual temperature difference in the recording room is about 8℃, and the daily temperature difference is less than 0.3℃.

### 2.1.2 Beijing Geomagnetic Observatory (IAGA code BJI)

The Beijing Geomagnetic Observatory, started construction in 1952, is located south of Baijiatuan Village in the western suburbs of Beijing (latitude 40.0, longitude 116.2), and was fully completed in 1954. In 1955, the observatory began to install observation instruments and started trial observation. In 1957, the Beijing Geomagnetic Observatory officially undertook the task of observation. The geology foundation of the observatory and surrounding area is composed of diluvial gravel layers, with Ordovician limestone buried more than 54 meters below, and the upper part is covered with layers of sand, gravel, and pebbles. Within the observation area, the magnetic field gradient is less than 1nT per meter. Both the absolute measurement house and the relative recording room are constructed with weak magnetic materials. The measurement house is a wooden and stone structure on the ground floor. The recording room is a semi-underground wooden and stone structure, with an annual temperature difference of 17ºC and a daily temperature difference of less than 0.3ºC. However, since 1995, with the acceleration of urbanization, the observation environment of the Beijing Geomagnetic Observatory has gradually been affected by surrounding construction projects. After 2012, with the opening of new subway lines, the interference to the observation environment has become increasingly severe.

### 2.1.3 Guangzhou Observatory (IAGA code GZH)

The Guangzhou Observatory (IAGA code GZH) was started construction in 1954, located 10 kilometers southeast of Guangzhou in Shiliugang (latitude 23.1, longitude 113.3). The observatory was completed in 1955 and equipped with geomagnetic observation instruments the same year. In 1958, the Guangzhou Observatory officially began geomagnetic observations. The geology foundation of the observatory and surrounding area is composed of sandstone, with a homogeneou magnetic field distribution and a magnetic gradient of less than 1nT/meter. Both the absolute measurement house and the relative recording room are ground-level buildings, constructed with weak magnetic materials such as limestone and wood. The daily temperature variation in the recording room is less than 0.2°C, and the annual temperature variation is about 17°C.

However, due to the construction of the Guangzhou rail transit and the South China Expressway, it was inevitable that interference would occur with geomagnetic observations. Therefore, the Guangzhou Observatory decided to relocate. The new site was chosen in Shendang Village, Liantang Town, Gaoyao District, Zhaoqing City, approximately 120 kilometers southwest of Guangzhou (latitude 23.0, longitude 112.5), and the site selection and positioning work was completed between 1996 and 1997. Construction of the new observatory began in November 2000 and was completed in December 2001. Thus,

there were no observation data available from 1996 to 2001. The bedrock at the site is composed of Carboniferous limestone. The magnetic field around the observatory is homogeneou, with a magnetic field gradient of less than 1nT per meter. Both the absolute measurement house and the relative recording room are constructed with weak magnetic reinforced concrete structures. The absolute measurement house is a ground-level building, while the recording room is a fully underground structure. The annual temperature difference in the recording room is less than 4°C, and the daily temperature difference is less than 0.1°C.

### 2.1.4 Wuhan Geomagnetic Observatory (IAGA code WHN)

The Wuhan Geomagnetic Observatory (IAGA code WHN) is located in the southeastern suburbs of Wuhan city, Hubei Province, in the town of Baozixie, Wuchang County (latitude 30.5, longitude 114.6). The construction of the observatory began in the spring of 1957 and was completed by December of the same year. In 1958, the instruments were installed and calibrated, and in 1959, the observatory officially began its observational. The observatory is built on a hill with a thick overburden, with a surface layer of yellow clay, where the distribution of the magnetic field gradient is relatively homogeneou, with a gradient of less than 1nT per meter. Both the absolute measurement house and the recording room are constructed with weak magnetic materials as ground-level buildings. The annual temperature difference in the recording room is about 28°C, and the daily temperature difference is generally less than 0.5°C.

Later, affected by the rapid development of township enterprises, it was relocated in 1999 to Jiufeng Village within the territory of Baozixia Town (114.5°E, 30.5°N). The geology foundation of the observatory and surrounding area is is mainly composed of Upper Triassic quartz sandstone, with thin layers of marl interbedded within. The magnetic field around the observatory is homogeneou, with a magnetic gradient less than 1nT per meter. The absolute measurement house and recording room are both constructed with weak-magnetic materials;the measurement house is a ground structure made of wood and stone, while the recording room is a ground structure made of copper, stone, and plastic, covered with a three-meter-thick soil layer. The indoor daily temperature difference is less than 0.1°C, and the annual temperature difference is less than 9.0°C.

Due to the influence of the subway,it was relocated to Tangchi Town, Yingcheng City, Hubei Province (113.3°E, 30.9°N) in 2017. The geology foundation of the observatory and surrounding area is made of limestone with a maximum thickness of 7.4 meters. The magnetic field gradient is less than 1 nT per meter. Both the absolute measurement house and the relative recording room are ground buildings made of weak-magnetic wooden and stone structures. The daily temperature difference in the recording room is about 0.1°C, and the annual temperature difference is about 15°C.

### 2.1.5 Sheshan magnetic observatory (IAGA code SSH)

The Sheshan magnetic observatory (IAGA code SSH), started construction in 1874, has maintained official records since its inception. Its predecessor was the Xujiahu Astronomical Observatory in Shanghai (1874-1908). Due to the construction of tram lines passing through Xujiahu in 1907, which inevitably interfered with geomagnetic recordings, the observatory  was

relocated to Lukjapu, Kunshan (1908—1933) in 1908. Later, because of the construction of the Beijing-Shanghai Railway, it was moved to its current location in Songjiang District, Sheshan, Shanghai (since 1932). The SSH Geomagnetic Observatory is located about 20 kilometers southwest of Shanghai in the Xisheshan area of Songjiang District (31.1°N, 121.2°E). With nearly 150 years of history, it is one of the longest-operating geomagnetic observatories in China and the world. The geology foundation of the observatory and surrounding area is mainly composed of medium to acidic andesite from the Upper

Jurassic to Lower Cretaceous periods. The magnetic field gradient is 2-3 nT per meter. Both the absolute measurement house and the relative  recording room are constructed using materials with  weak magnetic materials. The daily temperature difference in the recording room is less than 0.2℃, and the annual temperature difference is about 15℃.

### 2.1.6 Lhasa Observatory (IAGA code LSA)

The Lhasa Observatory (IAGA code LSA) is located on the west outskirts of Lhasa city (29.6 N, 91.0 E). Construction of the

observatory began in 1956, it was completed in 1957, and started formal observations in July of the same year (Zhou, 2013). Due to instrument failure, the observatory suspended its observation work from 1974 to 1982,and resuming in 1983. The Lhasa Observatory was the first highland geophysical observatory established in the world and was, at the time, the highest altitude geomagnetic observatory globally, with an elevation of 3,655 meters. The is located in the central part of the alluvial basin of the Lhasa River Valley in the suburbs of the city. The magnetic field gradient at the observation site is about 1nT

per meter, and is relatively homogeneou. Both the absolute measurement house and the recording room are non-magnetic ground buildings made of limestone, earth bricks, and wood structures. The daily temperature difference inside the recording room is less than 0.2℃, and the annual temperature difference is about 19℃.

### 2.1.7 Lhasa Observatory (IAGA code LSA)

The Lanzhou Geomagnetic Observatory (IAGA code LZH) is located at Liujiaping, north of the Yellow River in Lanzhou

City (36.1 N, 103.9 E). Construction began in 1956, was completed in 1957, geomagnetic instruments were installed in 1958, and trial observation were started. It officially commenced observation on January 1, 1959. The observatory is situated at Liujiaping, north of the Yellow River in Lanzhou City. The geology foundation of the observatory and surrounding area is mainly Quaternary loess layers. The ground magnetic field gradient is about 1nT per meter, and is relatively homogeneou. Both the  absolute measurement house and the relative recording room use non-magnetic or weakly magnetic stones and

earth bricks, as well as wooden structures for construction. The absolute measurement house is a ground-level building, while the recording room is an underground structure with a daily temperature difference of 0.3℃ and an annual temperature variation of no more than 10℃.

### 2.1.8 Lhasa Observatory (IAGA code LSA)

The Urumqi Observatory (IAGA code WMQ) was established in 1964 and officially began its observation in 1970. However,

due to the long-term instability of the instruments, the first geomagnetic observation report was not officially published until



1978. The observatory is located in the eastern suburbs of Urumqi, Xinjiang Uygur Autonomous Region, 10 kilometers from the city center in Shuimogou (43.8 N, 87.7 E). The geology foundation of the observatory and surrounding area is consists of Triassic sandstone and mudstone, with a Quaternary loess layer approximately 4 meters thick covering it. The ground magnetic field gradient is less than 1nT per meter. Both the absolute measurement house and the recording room are ground

buildings, constructed using weak magnetic soil bricks and a wooden stone structure. The annual temperature difference in the recording room is less than 8℃, and the daily temperature difference is less than 0.3℃.

Later, due to urban expansion and rapid socioeconomic development, the observation was significantly affected. Therefore, in 2012, the Urumqi Observatory  was relocated to Yuanhucun Village, Hutubi County, Changji Hui Autonomous Prefecture, about 80 kilometers northwest of Urumqi City (44.4 N, 86.9 E). The old observatory ceased operations in December 2012,

while the new observatory  officially began its observation on January 1, 2013. The geology foundation is covered with loess layers, and the observation field area has a homogeneou geomagnetic field with a ground magnetic gradient of less than 1nT per meter. The absolute measurement house and the recording room are both constructed with non-magnetic reinforced concrete and stone structures, with the absolute measurement house being a ground-level building and the recording room being a fully underground structure. The annual temperature difference in the recording room is less than 6°C, and the daily

temperature difference is less than 0.1°C.

Table 1 shows their position information and change conditions, which were published in the observatory's yearbook.

**Table 1** Changes in the location of geomagnetic observatories

| IAGA CODE | Location 1 | | Location 1 | | Location 1 | |
|---|---|---|---|---|---|---|
| | Geographic Latitude | Geographic Longitude | Geographic Latitude | Geographic Longitude | Geographic Latitude | Geographic Longitude |
| CNH | 43.8°N | 125.3°E | 44.0°N | 125.2°E | 44.1°N | 125.0°E |
| WMQ | 43.8°N | 87.7°E | 44.4N | 86.9°E | | |
| BJI | 40.0°N | 116.2°E | | | | |
| LZH | 36.1°N | 103.9°E | | | | |
| SSH | 31.1°N | 121.2°E | | | | |
| WHN | 30.5°N | 114.6°E | 30.5°N | 114.5°E | 30.9°N | 113.3°E |
| LSA | 29.6°N | 91.0°E | | | | |
| GZH | 23.1°N | 113.3°E | 23.0°N | 112.5°E | | |

## 2.2 Instrument Configuration


All eight observatories involved in this study are equipped with the absolute measurement house and the relative recording rooms. The absolute observation room is a ground-level building, while the relative recording room can be either ground-level



or underground. They are constructed using weak magnetic materials, providing an ideal environment for geomagnetic observations. Inside the absolute measurement house, the absolute measurement instruments are installed; in the relative recording room, the relative recording instruments are set up. The absolute measurement instruments is capable of measuring the absolute value of the geomagnetic field with high precision, but it cannot operate continuously; the relative recording instruments, on the other hand, can continuously record the changes in the geomagnetic field over long periods, although it cannot measure the absolute value of the geomagnetic field. Therefore, a combination of absolute observations and relative recordings is necessary to obtain long-term continuous absolute values of the geomagnetic field (Jankowsky and Sucksdorf 1996; Rasson 2007). In China, absolute measurement instruments are commonly used to determine the three independent elements of the geomagnetic field: magnetic declination (D), magnetic inclination (I), and total intensity (F). Absolute measurements are scheduled twice a week, usually on Mondays and Thursdays. The relative recording instruments are mainly used to measure the three elements of magnetic declination (D), horizontal intensity (H), and vertical intensity (Z).

Before 1975, the absolute observation instruments used on the eight observatories were produced by foreign manufacturers, including Askania, Schmidt, Smith, Cooke, Mating, QHM and other types of magnetometers, which were used to measure the absolute values of the D, H, and I components. Relative record relied on the Type 57 and Type 72 magnetometers developed independently in China. In 1975, China successfully developed the CJ6 geomagnetic theodolite, after which this instrument was used to absolute measurement of magnetic declination and horizontal intensity. At the same time, they used proton precession magnetometers of the CZM, DTZ, and other types to measure the F total field. Entering the 21st century, the China Earthquake Administration initiated the "Seismic Precursor Station (Network) Technical Renovation" project, vigorously promoting the digitization process of geomagnetic observatories, equipping them with GM-3 digital fluxgate magnetometers to continuously record changes in the geomagnetic field in the H, Z, and D components. In addition, they were equipped with CTM-DI fluxgate theodolites, combined with G856 proton precession magnetometers, to achieve absolute measurement of D, I, and F. Since 2007, the advancement of the "China Digital Seismological Observation Network" project has further updated and improved the observation instruments at the observatories, using MINGEO fluxgate theodolites and GSM-19T proton precession magnetometers for absolute measurement of D, I, and F, and equipped with GM4 fluxgate magnetometers for continuous relative recording of the D, H, and Z components. At the same time, the GSM-90F1 Overhauser magnetometer was introduced for continuous recording of the F total field.

The SSH Observatory, due to its early establishment, relied primarily on imported instruments for its early observation activities. In the field of absolute observations, it used magnetic instruments such as Uniflaire, Elliott, and Smith models to measure the three components (D, H and I) of the geomagnetic field. For relative records, it mainly used Toepfer-type recorders.

## 2.3 Data collection, organization, digitization, and standardization

Given that these valuable historical materials are distributed across different observatories or administrative departments, making it difficult to query and access data resources. Limited by the historical conditions at the time, storage methods vary,



with some records on paper carriers, others stored on outdated floppy disks and other storage media, and still others stored on modern hard drives. Due to the lack of unified standards and specifications, users find it difficult to obtain shared common data sources. Therefore, this study first aims to collect and organize these data from various scattered locations.

Data prior to 1985 mainly comes from annual reports of observatories, primarily in paper form. From 1985 to 2001, data was saved on floppy disks. Data from 2002 to 2006 was stored in Access databases in MDB format, while data from 2007 onwards has been saved in Oracle databases. After the initial collection process, we faced the more complex task of digital data processing. For historical materials recorded on paper carriers, we first needed to carefully scan them, storing the resulting digital images on hard drives for efficient use and long-term preservation. Given that paper materials have yellowed and become fragile over time, with reduced legibility, and some documents are handwritten, we found that the accuracy of recognition software for digital images was not as good as manual entry based on previous experience. Therefore, we manually entered this part of the data using pre-designed forms (Zhang et al., 2022b). For data stored on floppy disks, we needed to find compatible reading devices. Since floppy disk drives are rare on modern computers, we had to search everywhere and finally found a compatible drive at a second-hand electronics market, thus achieving the migration of data from the floppy disk to the hard drive. For data stored on modern hard drives, although technically easier to handle, we also needed to conduct a thorough inspection to ensure that the data was not damaged or lost. In addition, we also converted the collected data into a standardized format, unified the storage format, and imported all the data into an Oracle database for unified management, long-term preservation, and use. The data we collected covers the absolute hourly mean values (AHMV) of the X, Y, Z components from the following observatories: CNH Observatory (1957-2022), BJI Observatory (1957-2022), LSA Observatory (1957-2022, but no observation data from 1974 to 1982 due to instrument failure), SSH Observatory (1933-2019, no observation from 2020-2022 due to renovation of the observation room), GZH Observatory (1958-2022, no observation data from 1996-2001 due to observatory relocation), WMQ Observatory (1978-2022), LZH Observatory (1959-2022), and WHN Observatory (1959-2022).

## 3 Quality Inspection and Processing

Quality inspection and preprocessing of observational data are critical steps for conducting data analysis and applications. This study first performs a preliminary examination of the time series data through visual inspection to identify and eliminate obvious errors due to measurement mistakes, data entry omissions, or equipment failures. Subsequently, the tolerance test (Zhang et al., 2022b; Korte et al., 2009) is used to further identify and remove outliers from the data. After completing these steps, the "Multi-scale Wavelet Analysis Step Detection and Signal Correction Algorithm(MWASDSC)" (Zhang et al., 2023) is applied for further detection and processing of the data, primarily aimed at detecting and correcting jump phenomena (Chulliat and Telali, 2007) in the data, which are often caused by replacing absolute instruments, changing standard observation pillars, or relocating observatories. The MWASDSC is as follows: (1) Low-pass filtering: Apply low-pass filtering to the original signal to eliminate high-frequency interference and obtain a smoother low-pass filtered signal; (2)



Moving average processing: Perform moving average processing on the low-pass filtered signal based on different numbers of moving average points, resulting in a series of moving average signals. These signals reflect data characteristics at

different scales. (3) Constructing a dataset and wavelet transform: Based on the moving average signals obtained in (2), construct a dataset and choose the largest scale moving average signal to build a ramp function as the wavelet function. Perform a wavelet transform on it to obtain wavelet transform coefficients. At the same time, remove the largest scale moving average signal from the dataset; (4) Step jump point detection: Use the position of the maximum wavelet transform coefficient as the predetermined step jump point, and determine the optimal step jump point position and step height through

a search algorithm; (5) Enclose the area and repeat the search: Enclose the area that spans one ramp width before and after the optimal step jump point, repeat step (4), and continue to search for the optimal step jump point position and step height from the remaining area. (6) Constructing wavelet functions at different scales: Repeat steps (3) to (5), construct wavelet functions at different scales in sequence, and use wavelet analysis to pick up step information at different scales, including step jump point positions and step heights. When repeating step (3), look for the wavelet transform coefficients of the largest

scale moving average signal in the remaining scales moving average signals. (7) Statistics and correction: Statistically analyze all step information found by the wavelet functions at all scales. If three or more scales can identify the same step, it is preliminarily determined as a step that needs correction. Then, combine with the Kp index to investigate. If the Kp index is less than 5, it is officially determined as a step that needs correction (Kp index greater than 5 is usually large magnetic disturbance, which can also produce step effects). Based on the positions of the jump points and step heights of all confirmed

steps, correct the original signal in turn to obtain the corrected signal. (8) Iterative processing: Take the corrected signal in (7) as the original signal for the next round of signal processing, repeat steps (1) to (7), and search for and correct the remaining steps again.

This study successfully achieved quality inspection and processing of time series data through the aforementioned steps. The preliminary review and tolerance test effectively eliminated obvious error values and spike anomalies from the data. The

MWASDSC can identify and correct the jump in the data. Figure 2 shows the comparison of the AHMV before and after correction, with the corrected data showing significant improvement in continuity and reliability.

The analysis of the corrected data shows that each component has a significant year-to-year variation, which over a considerable period of time manifests as a monotonic increase or decrease. For the X component, there was a general trend of monotonic increase before 1962, and a monotonic decrease for the subsequent 60 years. The SSH observatory, due to its

longer observation, can more clearly exhibit this characteristic of change.The Y-component time series from the WMQ, LZH, and LSA observatories in the western region show a high degree of consistency in their trends. From the start of the observation, these three observatories exhibited an overall upward trend in the Y-component, reaching their respective peaks in 1997, 1995, and 1999, after which they transitioned into a downward trend until the present. At the same time, the other five observatories have shown a general downward trend since the beginning of observations. Additionally, all eight observatories

displayed varying degrees of upward convexity in the Y-component between 1987 and 2008. As for the Z-component, the CNH, BJI, LZH, and LSA observatories have shown a consistent year-on-year decline since observations began, reaching



minimum values around 1973, followed by an overall upward trend. The SSH observatory showed a slow upward trend before 1955, began a slow decline afterwards, reached its lowest point in 1973, and has been on an upward trend ever since.The remaining three observatories have roughly exhibited an upward trend since observations began.

Furthermore, in the time series curve of the AHMV, it is clear that the AHMV contain a rich variety of external field variation components. Since the external field originates from the ionosphere and magnetosphere current systems outside the Earth, the complexity and variability of these currents result in a diverse and complexly changing magnetic field morphology. It is due to this origin that the spatial distribution of the varying magnetic field is extensive, with a global scale and long-distance spatial correlation. This characteristic is particularly prominent in the X component, for example, during March 13th to 15th,

1989, all eight observatories recorded significant geomagnetic storm, and a noticeable decrease exhibited in the X component, with different observatories exhibiting varying magnitude of change.

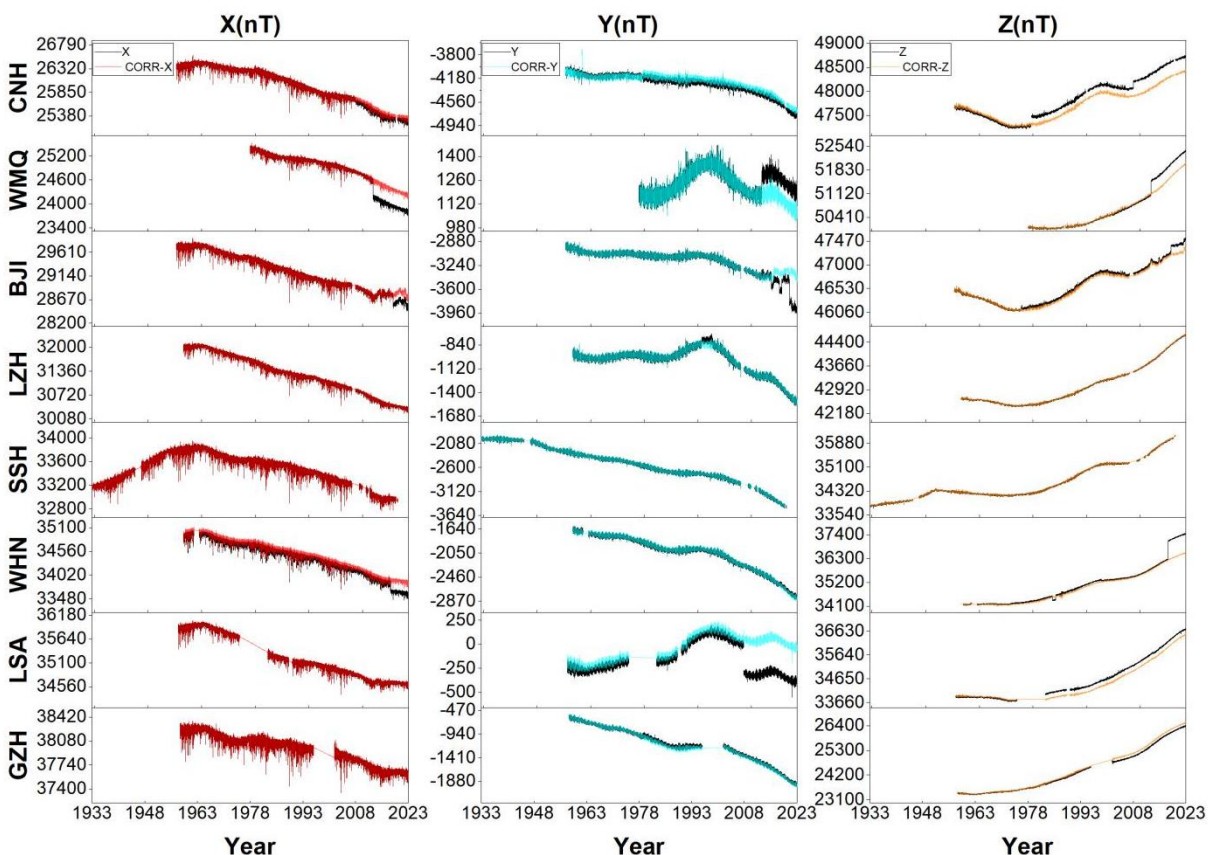

**Figure 2: Comparison of the AHMV before and after correction at the eight observatories**



## 4 Validation of corrected data

Validating the corrected data is an important means to evaluate data quality. The content of validation includes the accuracy of diurnal variation records and the stability of long-term changes (Curto and Marsal, 2007).

### 4.1 Validation of the accuracy of diurnal variation

Typically, the accuracy of daily variations is tested by comparing the consistency of daily variation data from neighboring observatories (Korte et al., 2007; Reda, 2011; Linthe, 2013). Since these eight observatories were among the earliest

established in China, there was a lack of data from nearby observatories for comparison in the early days. In this study, we adopted a strategy of cross-validation using data from the eight observatories (Curto and Marsal, 2007). Specifically, We used Geomagnetic Data Processing Software (Zhang et al., 2016) to conduct consistency comparative analysis of the diurnal variation patterns of these eight observatories on daily and monthly timescales.

The comparison results show that the data from the eight observatories, after correction, exhibit a high degree of consistency

in their daily variation patterns, with changes sometimes gentle and sometimes intense. Among these, one type of variation is gentle and periodic, cycling with a period of about 24 hours, with the magnetic field changes during the day being significantly greater than those at night. The other type of variation occurs occasionally, is turbulent and irregular, and disappears after a period of time (Xu, 2009). This phenomenon is mainly attributed to their origins from different current systems. The gentle and periodic variations originate from relatively stable current systems in the ionosphere (Pedatella et al.,

2011; Yamazaki et al., 2016), while the intense and irregular variations originate from various transient current systems formed by solar particle flows in the magnetosphere and ionosphere (Campbell, 2003). In most cases, regular magnetic field changes are superimposed with some irregular changes. Taking the records from February 1986 as an example, as can be clearly observed from Figure 3, the daily records of each observatories include both regular periodic quiet daily variations and complex disturbed variations, especially during the magnetic storm from February 7th to 9th (As shown between the two

vertical dashed lines in Fig. 3), where all observatories' three components showed a consistent intense disturbed response.

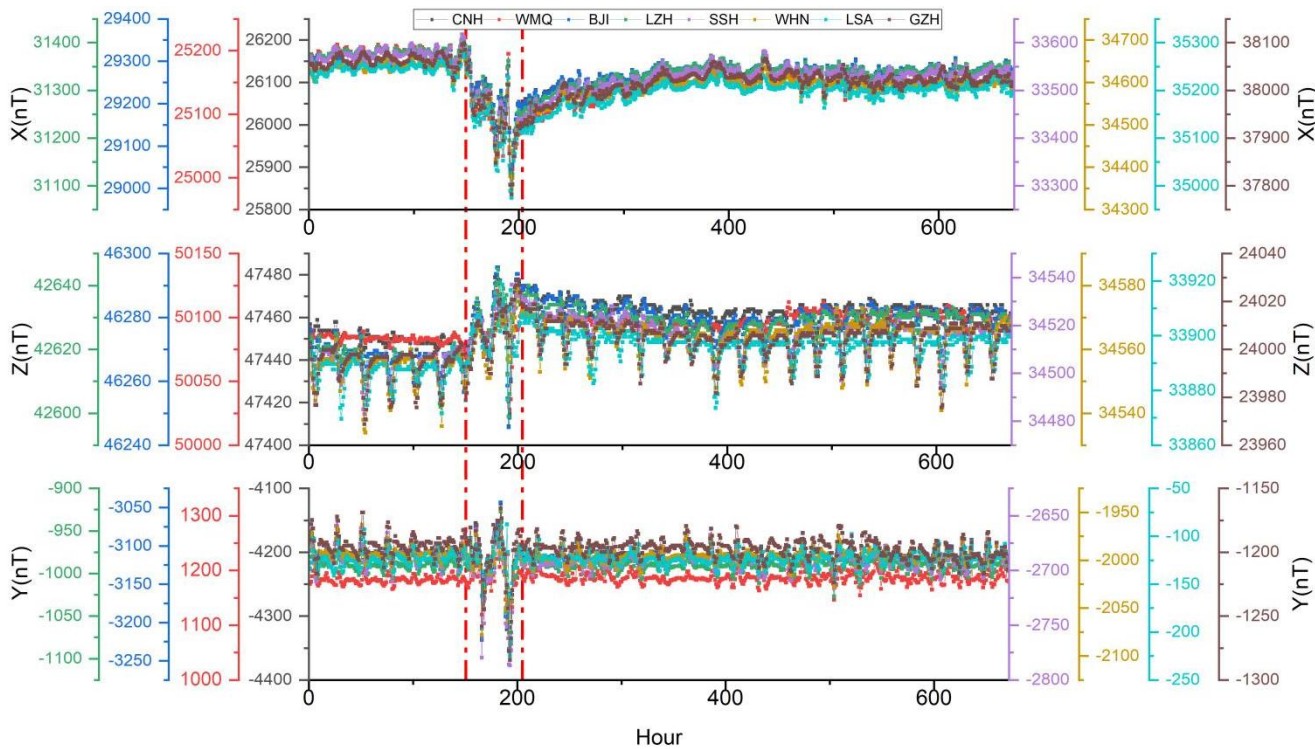

**Figure 3: Time series curve of AHMV of X, Y and Z components from eight observatories in February 1986**

## 4.2 Validation of the stability of long-term changes

Comparing the processed data with the prediction data from mature and reliable model can serve as an important method for
evaluating the long-term stability of data (Kudin et al., 2021), and can also be used to test the model's adaptability to ground observation data (Finlay et al., 2016). The comparison model selected for this study is the International Geomagnetic Reference Field (IGRF) model. The IGRF model is a mathematical model of the internal geomagnetic field widely recognized by the international academic community, constructed based on candidate models developed by multiple research teams. It is mainly describing the long-wavelength components of the main magnetic field. The IGRF model has also been
widely accepted and applied in practical applications. The latest version of the IGRF model can be obtained from the website (http://www.ngdc.noaa.gov/IAGA/vmod/) and supports online direct calculation of the seven elements of the geomagnetic field (X, Y, Z, F, D, I and H) and their secular variations at specified coordinates and dates. The update cycle of the IGRF model is every five years, and the coefficients of the 14th generation IGRF model were finally determined by IAGA working group in November 2024.
To exclude interference from external fields, this study selected mean data from the midnight period from eight observatories to calculate the annual mean (He et al., 2019). The midnight mean data was obtained by calculating the arithmetic average of the AHMV from 00:00 to 03:00 local time at each observatory. During this period, the observatories are located on the

Earth's night side, where the geomagnetic field is relatively stable, effectively avoiding interference from the ionospheric current system and some of the magnetospheric current system. In addition, the midnight period also avoids interference caused by a large amount of human activity. Furthermore, to eliminate the influence of the lithospheric magnetic field, we subtracted their respective initial year's annual mean from the annual mean of each observatory and the IGRF model every year (Figure 4).



**Figure 4: Comparison of annual mean of the observatories with annual mean calculated by the IGRF14 model**

As observed, it is evident that the three components from observatories are highly consistent with the long-term variation trends shown by IGRF model. This indicates that the corrected data can accurately reflect the long-term characteristics of the geomagnetic field in the China. Specifically, both the X-component showed an upward trend before 1962, which then transitioned to a monotonic downward trend; for the Y-component, the WMQ, LZH, and LSA observatories in the western region exhibited a high degree of synchronicity in their trends. Before the late 1990s, the overall trend was upward, followed by a shift to a downward trend until the present. The other five observatories generally showed a downward trend; regarding



the Z-component, the CNH, BJI, LZH, and LSA observatories showed a slight annual decrease before 1973, after which the overall trend became upward; the SSH observatory showed a slow increase before 1955, followed by a decrease until reaching a minimum in 1973, after which it has been on an upward trend. The other three observatories have consistently shown an

upward trend. Additionally, for the BJI observatory, the deviation between the observed and model for the X and Y components after 2000, and for the Z component after 2012, has shown an increasing trend. It is speculated that this phenomenon may be due to the increasing interference in the observation environment.

To further quantitatively analyze the differences between the observation and model, this study used formula (1) to calculate the differences between the observatory values and model values obtained above, as well as the standard deviation of the

differences. The comparison of the differences is shown in Figure 5, and the statistical results of the standard deviation of the differences are shown in Table 2.

$$\Delta B = B_{\mathrm{OBS}} - B_{IGRF}$$

$$S = \sqrt{\frac{\sum_{i=1}^{n}(\Delta B_i - \overline{\Delta B_i})^2}{n}} \tag{1}$$

In the formula, $B_{OBS}$ represents the annual mean of the observatory, $B_{IGRF}$ represents the annual mean of the IGRF model, $\Delta B$ represents the difference between the annual mean of the observatory and the value of the IGRF model, $\overline{\Delta B_i}$ represents

the mean of the differences, $S$ represents the standard deviation of the differences, $n$ represents the total number of years, and $i$ represents the $i$-th year.

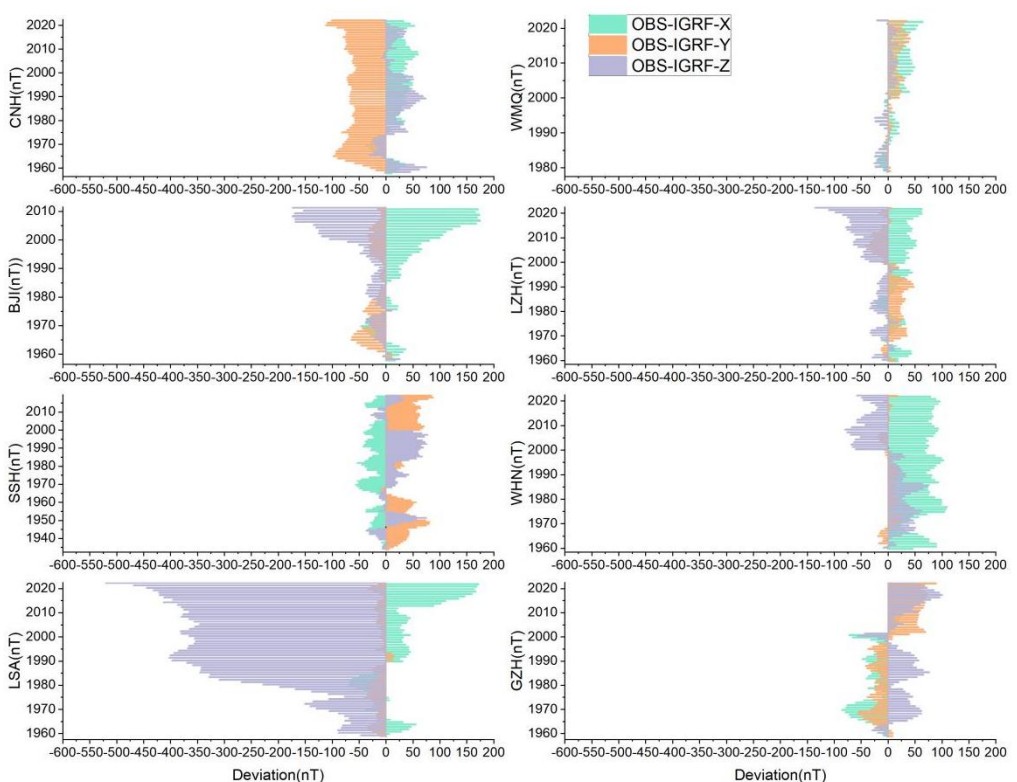

**Figure 5: Difference between observatory values and IGRF model values**

Although the long-term trend of the observatories are consistent with the IGRF model values, it can be seen from Figure 5 that there are certain differences between them. The difference in the Z component of the LSA observatory is the largest, with the maximum difference exceeding 500 nT and the average difference being about 280 nT. Following that, the differences in the X and Z components of the BJI have significantly increased since 2000, with the maximum difference approaching 200 nT, which may be related to the gradual deterioration of the surrounding observation environment (Zhang, 2018). The differences

at other observatories are basically all below 100 nT, or even smaller. The error statistics in Table 2 also show that, apart from the slightly larger errors in the X and Z components of LSA and BJI, the root mean square errors of the X, Y, and Z components at the other observatories are all within 50 nT. Preliminary analysis suggests that the reasons for the differences mainly involve two aspects: first, factors such as the uneven distribution of observatories, the degree of participation of observation data, data quality, measurement errors, boundary effects, truncation errors, and extension errors, etc., introduce

errors of different magnitudes during the IGRF modeling process (Lowes, 2000; Nie et al., 2017). Second, although this study has tried to exclude the interference of external fields and the lithospheric magnetic field as much as possible, the annual mean of the observatories may still contain other components besides the long-wavelength component of the main magnetic field that the IGRF model mainly reflects.



**Table 2** Statistical results of standard deviations for the differences between observatories values and IGRF model values

|        | CNH | WMQ | BJI | LZH | SSH | WHN | LSA | GZH |
|--------|-----|-----|-----|-----|-----|-----|-----|-----|
| **X(nT)** | 24  | 22  | 64  | 22  | 18  | 22  | 59  | 31  |
| **Y(nT)** | 19  | 14  | 17  | 23  | 23  | 12  | 12  | 43  |
| **Z(nT)** | 27  | 13  | 57  | 32  | 30  | 45  | 143 | 34  |


By analyzing the differences between them, the overall accuracy of the IGRF model in China can also be assessed. Although there are certain differences, considering that the global estimation accuracy range of the IGRF model is between 5 and 300 nT (Lowes, 2000; Wang, 2003; Nie et al., 2017; Lowes, 2022), the model is capable of accurately depicting the spatial distribution of the main magnetic field in China. In scientific research and practical applications of the geomagnetic field, it is
crucial to know the accuracy of the model in specific areas.

**5 Analysis Application**

The application scope of geomagnetic observation data is very extensive, as mentioned in the preface. In this section, we will introduce its applications in the study of Sq variations and geomagnetic jerks.

**5.1 Analysis of the Characteristics of Sq Variation in the China**

In the calm variations, the solar quiet daily variation (Sq) has attracted attention due to its theoretical significance and practical value. Sq mainly originates from the ionospheric dynamo current system (Vichare et al., 2017) and is the most significant variation component in the mid-low latitude regions during the quiet period of the geomagnetic field. This section will briefly analyze the characteristics of Sq variation in China using the corrected hourly mean data.

The data from the five calmest days of the geomagnetic field in February 1986 were selected, and the average Sq for that
month was obtained through time series superposition. The international geomagnetic quiet day data comes from the World Data Center (https://wdc.kugi.kyoto-u.ac.jp/). As shown in Figure 6, the horizontal axis represents local time. From the figure, it can be seen that the variations of the three components of Sq at various geomagnetic observatories show the characteristic of large daytime variation and small nighttime variation. This is because the Sq current system has a counterclockwise daytime current vortex in the mid-low latitude part of the Northern Hemisphere (the center position of the
maximum current intensity), and the center of this current vortex is located at approximately latitude 30°, while the current intensity at night is very weak (Zhao et al., 2014). It can also be seen from the figure that the X component reaches a significant extreme value before noon, the Y component shows a reverse extreme value around local noon, and the Z component has a characteristic of a minimum value appearing around noon. These characteristics are closely related to the current vortex center of the Sq current system being around local time 12 o'clock. In addition, the variation of Sq in the mid-
low latitude regions is mainly influenced by two factors: latitude and local time (Xu, 2009; Feldstein and Zaitzev, 1968),





which is also clearly reflected in Figure 6. Sq shows a significant dependence on local time. Comparing observatories with the same latitude but different longitudes, such as CNH (43.8 N, 125.3 E) and WMQ (43.8 N, 87.7 E), it can be seen that their Sq variation patterns are very similar, almost independent of longitude, showing characteristics of smooth changes with local time. The latitude variation characteristics of Sq are mainly manifested as the phase of the X component approximately

reverses at latitude 30 degrees; the Y and Z components do not show a reversal phenomenon, because the Y and Z components usually show variations in the equatorial region (Campbell, 2003; Amory-Mazaudier, 2009; Yamazaki and Maute, 2017). In addition, the changes in the X and Z components show approximate symmetry around local noon, while the Y component shows asymmetry. These variation characteristics are consistent with the temporal and spatial variation patterns of Sq in the mid-low latitude regions during the winter solstice month of the Lloyd seasons.

We also analyzed the amplitude characteristics of Sq. The bar chart in the figure shows the daily variation amplitude of the three components at each observatories. The daily variation amplitude of the Y component is the largest at the GZH observatory with the lowest latitude. The daily variation amplitude of the Z component shows that for observatories with similar longitudes, the lower the latitude, the larger the daily variation amplitude. The variation amplitude of the X component does not show a clear regularity.

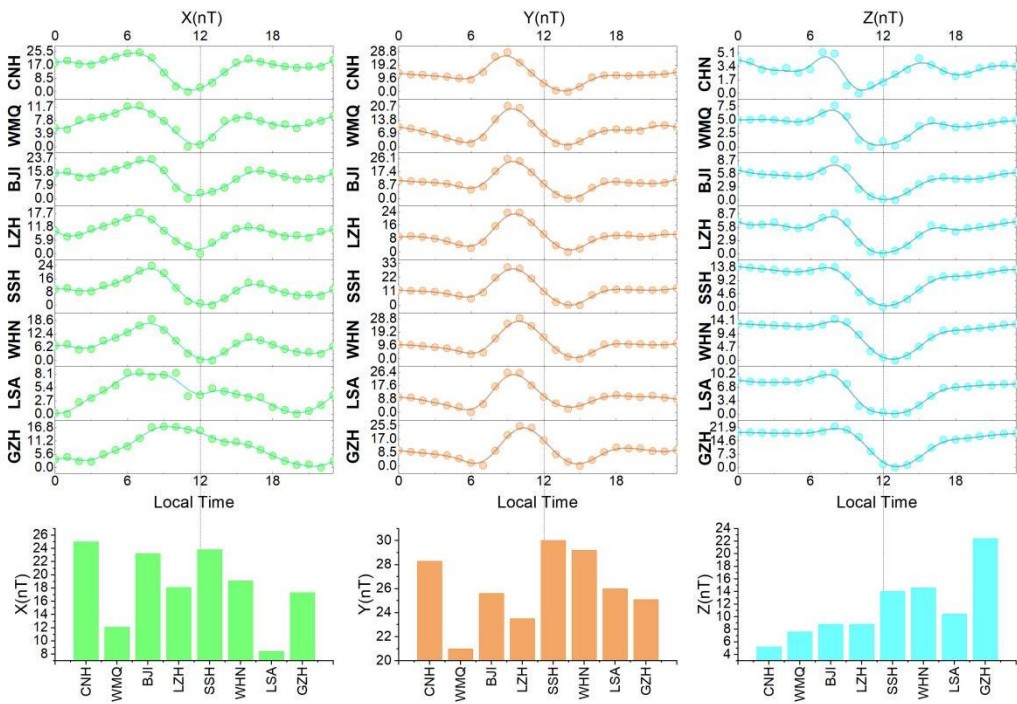


**Figure 6: Sq in February 1986**



## 5.2 Geomagnetic jerk analysis

Geomagnetic jerks, which refer to rapid changes in the geomagnetic field occurring over short periods, are characterized by abrupt changes in the secular variations of geomagnetic field elements, showing a "∨ " or "∧ " shape. The academic

community generally agrees that the origin of geomagnetic jerks can be traced back to the Earth's core, and these sudden changes are usually related to the flow changes in the Earth's core (Malin, 1982; Alexandrescu, 1995; Mandea, 2010; Chulliat, 2010; Qamili, 2013;Ou et al., 2016). Although geomagnetic jerks have received extensive attention and research in the past forty years (Chulliat, 2010; Brown, 2013; Feng et al., 2019; Kang, 2020; Bai, 2023), there are still many unsolved mysteries in the study of their origin mechanisms and change characteristics. The purpose of this section is to analyze and

discuss the geomagnetic jerks using the acquired long-term observation data in this study.

The most direct method for detecting geomagnetic jerks is to identify V-shaped or Λ-shaped changes in the secular variation (SV) of a specific component of the geomagnetic field, after excluding the external field components (Soloviev, 2017). This part of the study still selects annual mean values calculated from AHMV based on the midnight period, effectively avoiding interference from external magnetic fields (variations in the ionosphere and magnetosphere) and human activities. The

general view is that the X and Z components are more susceptible to the effects of external fields, while the Y component is least affected by external fields, making the jerk's manifestation most significant on the Y component (Mandea et al., 2010; Duka, 2012; Kang et al., 2020). Based on this, this study focuses on analyzing jerks using the Y component.

Next, we obtain SV value by calculating the first-order difference of the annual mean of Y component (Mandea et al., 2000; Nahayo et al., 2018; Kang et al., 2020). By taking the difference, we can eliminate the constant crustal magnetic field

component (Bloxham and Jackson, 1992; Feng et al., 2018). The SV is calculated as:

$$SV_Y = (Y(year) - Y(year-1))/1 \qquad (2)$$

In the formula, $SV_Y$ represents the SV value of the Y component, $Y(year)$ represents the current annual mean, and $Y(year-1)$ represents the annual mean of the previous year.

In Figure 7, the black dots represent the SV values of the Y component at each observatory. The blank areas in the data are

due to the lack of observation data during the corresponding time periods. Among them, the observation environment at the BJI observatory has been severely disturbed since 2012, so the data from 2012 and onwards were not included in the analysis. The GZH observatory was relocated and reconstructed due to environmental interference between 1996 and 2001, resulting in no observation data during that period.

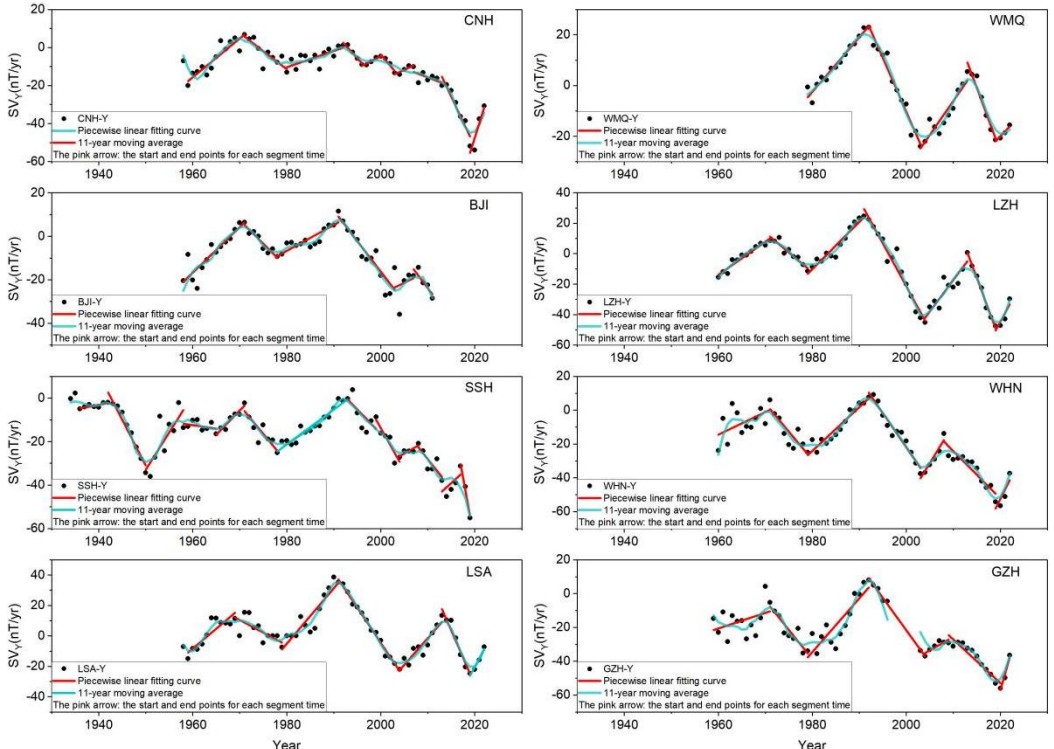

**Figure 7: The SV of the annual Y series from the eight observatories**

To quantitatively characterize the Jerk features, this study calculated the SA values before and after each jerk and the occurrence time of the jerk. The estimation of SA values is based on piecewise linear fitting of the time series, using the method proposed by Torta et al. (2015) and Kotzé (2011). The specific steps are as follows: First, to minimize variations related to the solar cycle 11, a 11-year moving average is applied to the first-order differences of the annual means (Alexandrescu, 1997; Korte, 2009). By observing the trend and turning points of the moving average curve, the starting and ending points of each time window are determined, and then piecewise linear fitting is performed. Figure 7 well depicts this process. The intersection time of two adjacent fitted linear segments in the figure represents the moment of jerk occurrence. The slope of the fitted straight line represents the SA value (Mandea et al., 2000; Pinheiro et al., 2011; Brown, 2013; Bai et al., 2023). The intensity of the Jerk can be quantified by its amplitude, which is the difference in SA values before and after each jerk, A, with its fitting formula as.

$$SV_Y(t_i) = a_1 t_i + b_1, \quad t_i \leq t_0$$
$$SV_Y(t_i) = a_2 t_i + b_2, \quad t_i \geq t_0 \tag{3}$$
$$A = a_2 - a_1$$

In the formula, $SV_Y(t_i)$ represents the SV value of Y component in $t_i$ year , $a_2$ indicates the SA value after the jerk, and $a_1$ is the SA value before the jerk.





As shown in Figure 7, several jerks appeared in the SV of the Y components at every observatory. The most notable events
occurred in 1969, 1979, 1991, and 2003 (Mandea and Olsen, 2007), which were clearly reflected in the SV at each
observatory. The 2003 jerk occurred distinctly at all observatories, further confirming Tozzi's view that this event was also
global (Tozzi et al., 2009). The Jerk in 2007 (Kotzé, 2011) and 2013 (Torta et al., 2015; Chulliat et al., 2015; Finlay et al.,
2016) were recorded by five observatories, with three western observatories not recording the event in 2007. The Jerk event
in 1999 was only recorded by two observatories, CNH and SSH, and its morphological characteristics were not very clear,
which may indeed be a local jerk (Pinheiro et al., 2011). These phenomena further illustrate that some jerk events have
global extension characteristics, while some jerks are local events (Chulliat et al., 2010). Furthermore, it is surprisingly that
the latest jerk event in 2019 was recorded by various observatories, which further confirms the prediction by Duan (2020)
that a new geomagnetic rapid change may occur between 2020–2021. Duan et al.'s research predicts, based on the
correspondence between LOD changes and geomagnetic jerks, that a new geomagnetic jerk may occur between 2020 and
2021.This phenomenon deserves further attention.

The 1949 Jerk event (Alexandrescu et al., 1996) is clearly recorded in the Y component of SSH SV, with very clear
characteristics of the Jerk. However, due to its early occurrence, there was only one observatory in China at that time,
lacking data from other observatories to corroborate, thus it cannot be determined whether it was a national event or an
isolated regional event. Additionally, events with similar jerk characteristics were recorded by the CNH observatory in 1996,
and by the SSH observatory in 1942, 1958, and 1965. Among these, the 1942 event (Mandea et al., 2000; Duka et al., 2012),
and the 1958 event were only detected in certain observatories of the world (Lutephy, 2018). These jerks may be real or
could be related to the length of geomagnetic records, data quality, etc., especially for data from the early initial of research,
which requires more early data for confirmation.

Table 3 shows the occurrence times and amplitude A of each jerk event in the Y component of various geomagnetic
observatories. Positive amplitude indicates that the secular variation exhibits a V-shaped feature, while negative amplitude
suggests that the secular variation exhibits a Λ-shaped feature. Blank spaces in the table indicate that no jerk was detected
during the corresponding time period. The first column of the table lists the widely recognized occurrence times of
geomagnetic jerks.

**Table 3:** The occurrence times and Amplitude of Jerks

| Jerk event | Year/Amplitude | CNH | WMQ | BJI | LZH | SSH | WHN | LSA | GZH |
|---|---|---|---|---|---|---|---|---|---|
| 1942 | Time (year) | | | | | 1942 | | | |
| | Amplitude (nT) | | | | | -4.6 | | | |
| 1950 | Time (year) | | | | | 1950 | | | |
| | Amplitude (nT) | | | | | 7.6 | | | |
| 1958 | Time (year) | | | | | 1958 | | | |
| | Amplitude (nT) | | | | | -3.7 | | | |





| 1965 | Time (year) | | | | | 1965 | | | |
| | Amplitude (nT) | | | | | 2.5 | | | |
| 1969 | Time (year) | 1971 | | 1970 | 1971 | 1971 | 1971 | 1969 | 1971 |
| | Amplitude (nT) | -3.9 | | -4.3 | -4.8 | -4.8 | -4.7 | -4.3 | -4.4 |
| 1979 | Time (year) | 1980 | | 1978 | 1979 | 1978 | 1979 | 1979 | 1979 |
| | Amplitude (nT) | 2.8 | | 3.3 | 5.7 | 4.2 | 5.9 | 5.3 | 6.6 |
| 1991 | Time (year) | 1992 | 1991 | 1991 | 1991 | 1993 | 1992 | 1991 | 1992 |
| | Amplitude (nT) | -3.6 | -6.4 | -4.0 | -8.8 | -3.8 | -6.6 | -8.3 | -7.0 |
| 1996 | Time (year) | 1996 | | | | | | | |
| | Amplitude (nT) | 4.0 | | | | | | | |
| 1999 | Time (year) | 1999 | | | | 1999 | | | |
| | Amplitude (nT) | -4.2 | | | | -1.6 | | | |
| 2003 | Time (year) | 2003 | 2003 | 2003 | 2004 | 2004 | 2003 | 2004 | 2004 |
| | Amplitude (nT) | 4.0 | 6.9 | 3.9 | 9.8 | 5.1 | 8.6 | 8.1 | 5.4 |
| 2007 | Time (year) | 2007 | | 2007 | | 2008 | 2008 | | 2009 |
| | Amplitude (nT) | -2.1 | | -4.0 | | -4.0 | -7.3 | | -4.2 |
| 2013 | Time (year) | 2013 | 2013 | | 2013 | 2013 | | 2013 | |
| | Amplitude (nT) | -4.3 | -7.7 | | -12.4 | 4.7 | | -10.6 | |
| 2019 | Time (year) | 2019 | 2019 | | 2019 | 2019 | 2019 | 2019 | 2019 |
| | Amplitude (nT) | 13.2 | 6.9 | | 14.1 | -13.9 | 8.4 | 13.0 | 12.3 |


As shown in Table 3, the geomagnetic jerks exhibit significant regional differences and asynchronous occurrence times (Torta et al., 2015). For the same jerk event, there are considerable differences in the amplitude recorded by different observatories. Overall, the jerk amplitudes recorded by observatories in the central and western observatories are greater than those in the eastern observatories, except for the jerk event in 2019.

For different jerk events, the amplitude at the same observatory also varies significantly, ranging from about 2nT yr-2 to several tens of nT yr-2. Notably, for jerks with more pronounced morphological features, the absolute value of their amplitude exceeds 4nT yr-2 (Soloviev, 2017). The occurrence times of the jerks do not show any clear pattern in the eight observatories studied; the time difference for the same jerk recorded at different observatories can reach up to 2 years. This result is consistent with the conclusions of other scholars, who noted that the time discrepancy of the same event at different

observatories can reach 2 to 3 years (Mandea et al., 2010; Brown et al., 2013; Morozova et al., 2014; Soloviev, 2017). The intervals between jerks are often 4 years, 6 years, 8 years, and 12 years, which seems to be correlated with the 6-year and

8.6-year oscillation periods of Earth's rotation diurnal length variations (Ma, 2004; Holme and De Viron, 2013; Lutephy, 2018; Duan and Huang, 2020; Gvishiani and Soloviev, 2020).

## 6 Data availability

An integrated and quality-controlled historical datasets of eight pioneering geomagnetic observatories in China are available at: https://doi.org/10.5281/zenodo.14560950, 2024 (Zhang et al., 2024b). The data are provided in Txt format, including the observed AHMV files of the three components (X, Y and Z) at the eight observatories, and also the metadata files about the datasets.

## 7 Conclusion

Geomagnetic observatories play a crucial role in monitoring the spatiotemporal variations of the geomagnetic field, and their accumulated continuous observation data are particularly important for understanding the long-term evolution patterns of the geomagnetic field. Long time-series geomagnetic observation data have wide applications and significant value for scientific research and practical use.The rescue of geomagnetic historical data is a focal point of global scientific concern. This study conducted a salvage organization of the historical observation data from eight geomagnetic observatories (CNH, WMQ, BJI,

LZH, SSH, WHN, LSA,and GZH) from the official start of observations to 2022. This included collecting disperse stored observational materials, digitizing paper materials, unified database management, anomaly detection and processing, quality assessment, and application analysis. After the salvage, the historical data is easier to preserve and use over the long term, with significantly improved data quality. These datasets are of great value for improving historical geomagnetic field models (Korte, 2009), studying changing magnetic fields (including diurnal variations and magnetic storm events, etc.), the main

geomagnetic field, and its secular variations. However, it should be noted that after 2012, the data from the BJI observatory should be handled with caution due to severe interference in the observation.

In Chapter 5 of this study, we first applied the data to explore the changing characteristics of Sq in China. The analysis results indicate that the main features of Sq include significant changes during the day, weak changes at night, and the changing patterns also show dependence on latitude and local time. In addition, we also focus on the changing characteristics of jerks in the China.

The research results revealed several landmark jerks in 1969, 1979, 1991, and 2003 that commonly appeared at eight observatories, thus verifying the global characteristics of these events. In contrast, events in 1999, 2007, and 2013 exhibited local characteristics. For events in 1942 and 1958, since they were only recorded at one observatory, more long-term sequence data are needed for in-depth analysis and confirmation. Encouragingly, the jerk in 2019 was recorded at all observatories. For those jerks with significant morphological features, their amplitude absolute values all exceeded 4nT yr-2.

These findings further confirm the regional differences in geomagnetic jerks and the asynchronicity of the occurrence times of jerks, which is of significant scientific importance for understanding changes in the geomagnetic field.

Although this study reveals the spatiotemporal characteristics of Sq and jerks in China through analysis, given that the data samples used may lack sufficient representativeness and the analytical methods may have certain limitations, these findings may not fully and accurately reflect the actual situation in China. Therefore, subsequent research will collect and integrate more high-quality data from observatories for more detailed and in-depth analysis. At the same time, we will focus on improving the accuracy of the detection algorithm and delve into the physical mechanisms of geomagnetic jerks.

This study aims to provide this valuable dataset to the scientific community and the public, so that these data can be integrated with data from other sources to further explore the spatiotemporal evolution of the geomagnetic field and its physical mechanisms. It also hopes to inspire more scientific workers to engage in the protection of historical materials, with the aim of preserving these precious resources before they are destroyed or lost, and to further realize their scientific value.

**Author contributions**. SZ conducted data integration, quality inspection,, and analyzed application examples. SZ prepared the manuscript. CF was responsible for identifying and correcting data anomalies, created charts, and revised the manuscript. CC, GZ, JC, QZ, BW, ND, JL, SH, PG and HY is responsible for scanning and digitizing paper materials, as well as standardizing data in various formats. CC designed a set of Excel templates. JW developed the data quality comparison software. JW and CC established database management system

**Competing interests.** The contact author has declared that none of the authors has any competing interests.

**Acknowledgements**. We thank CNH, BJI, SSH, WMQ, LZH, LSA,GZH, WHN, and the reference room of the Institute of Geophysics, China Earthquake Administration for providing the valuable data resources. We sincerely thank all the technical personnel and scientists working on the geomagnetic observatories. We also thank World Data Center for Geomagnetism, Kyoto, for providing online International Geomagnetic Quiet Day.

**Financial support**. This work was funded by the National Key R&D Program of China (grant no. 2023YFC3007404), the National Natural Science of Foundation of China (grant no. 42374092).

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
