# Peer review of "Rescue, Integration, and Analytical Application of historical data from eight pioneering geomagnetic observatories in China"

_Earth System Science Data, 2025_

## Author Response (AR1)

**Author's response to reviews on ESSD-2025-6**

The authors are thankful to the reviewers for their valuable comments. These comments are very helpful for revising and improving our manuscript. All the modifications are as follows. Corresponding changes have been made in the revised manuscript and are marked with "track changes".

**Referee #1:**
**General Comments:**
It is obvious that the authors of this work have accomplished a very valuable and meaningful task. Scientists working on related research can utilize results of this work for in-depth studies. I strongly recommend publishing it as soon as possible.

**Thank you very much for your affirmation and valuable suggestions on our work.**

**Specific Comments:**
But for publication, the manuscript needs several minor modifications. The specific suggestions are as follows:
**Comment:** 1. The titles of Section 2.1.7 & 2.1.8 not match the content, and need to be modified.
**Response:** We have revised the titles to better reflect the content of these sections.

**Comment:** 2. Why was the data after 2022 not processed, and what's the reason? Please briefly explain in the article.
**Response:** The data after 2022 was not included because it was incomplete at the time of our analysis. We have added a brief explanation in Section 2.3 to address this issue.

**Comment:** 3. In Figure 6 on page 19, the unit of Local Time should be labeled.

We thank you for pointing out the missing unit for Local Time in Figure 6. We have now labeled the unit clearly in the revised figure. (Figure 7 in our revised manuscript).

**Referee #2:**
**General Comments:**
The manuscript presents digitized data from eight long-term geomagnetic observatories in China. Digitizing, preserving and archiving such data and making them available for research is a highly important task to ensure that previous scientific efforts and valuable sources of information about the long-term evolution of natural phenomena (in this case the geomagnetic field and its variations) does not get lost and remains available for present and future research. The manuscript provides a generally good description of the data sources and the presented data and is complemented by quality checks of the digitized data and two examples for applications. The dataset and manuscript certainly deserve publication.

**We sincerely appreciate your positive assessment of our manuscript and your recognition of the significance of digitizing and preserving geomagnetic data from Chinese observatories. Your detailed feedback has been invaluable in helping us enhance the quality and clarity of our work.**

**Specific Comments:**

I have several requests for additional information or clarification that in my opinion would make the manuscript even more useful as documentation of the newly available data. My main concern is the first point below, as it can be quite concerning and confusing for potential users of the data if data that should be the same but are obtained from different sources differ. The data are available as described and easy to read and use, but I also have some requests for clarification on it.

**Comment:** a) It should be clearly stated which data have been newly digitized, and for the overlapping part the data should be carefully checked against the hourly mean values already held at the World Data Center for Geomagnetism in Edinburgh (https://wdc.bgs.ac.uk/data.html , https://wdc-dataportal.bgs.ac.uk/) . The updated compilation from the eight observatories as presented here should be sent to the WDC so that that database also is up to date, and only one version of the data is around. I didn't check for all eight observatories, but, e.g., BJI data are available at the WDC from 1957 – 1994 with a short gap around 1879, and LZH data are available for several years between 1980 and 2019, so some of the data presented here in a more complete form are already available there.

**Response:** Thank you for providing valuable review comments and suggestions.

We adopted a completely manual input method to digitize the data from eight observatories prior to 1985.

We obtained the hourly mean values already held at the World Data Center for Geomagnetism in Edinburgh (WDC) and conducted a detailed comparison with our newly integrated data. We observed significant deficiencies in the continuity and completeness of the WDC data, and through our integration efforts, we successfully filled in the missing data for many blank years.

These contents have been supplemented in Section 2.3 of the revised manuscript.

**Comment:** b) The description of the observatories and their history is very valuable. For an easier assessment of the potential influence of re-locations, it would be useful to give the approximate distance between old and new location in km in addition to the coordinates. Is it true that in none of the cases of re-location observations at both locations to determine the resulting "jumps" in the data series have been done (it is not mentioned in any of the cases)? If this has been done it should be mentioned.

**Response:** Thank you for your positive comments and valuable suggestions.

We have added content including the distance between old and new locations, as well as the operating start and end years of each location in Table 1 of the revised version of the manuscript.

Regarding your question about whether observations were conducted at both locations during relocations to determine resulting "jumps" in the data series, after carefully reviewing relevant archives, we have located pertinent information in the

"Geomagnetic Observation Report" and a reference paper. We found that in most cases, such comparative observations were indeed implemented. Accordingly, we have elaborated on this point in the section 2.1 the revised version of the manuscript.

**Comment:** c) The first sentence of section 2.1.5 is confusing – the mentioned construction start in 1874 was not in Sheshan, but in Xujiahu, right? What was probably meant to say with the first sentence is that Sheshan (with predecessors) has the longest uninterrupted data series? Please re-formulate for clarity.

**Response:** Thank you for bringing this to our attention. The first sentence of section 2.1.5 was confusing. The construction that started in 1874 was indeed in Xujiahu, not Sheshan. What we intended to convey was that, despite the construction starting in Xujiahu, the Sheshan observatory (including its predecessors) has the longest uninterrupted data series. We have revised the first sentence of section 2.1.5 to eliminate the confusion and clearly state this point.

**Comment:** d) I suggest to put the sentence mentioning Table 1 and the table itself at the beginning of section 2.1, i.e. before 2.1.1. At the moment the sentence seems to belong to section 2.1.8, although it refers to all the observatories described in 2.1. It could also be informative to expand the table to include information about the available data, i.e. start of the data series and gaps.

**Response:** Thank you for your helpful suggestion. We agree that moving them to the beginning of section 2.1 would improve the flow and clarity of the text, as the table provides an overview of all the observatories discussed in section 2.1. We have revised the manuscript accordingly. In addition, we have also expanded Table 1 by adding information about the available data.

**Comment:** e) The description of the used instruments in section 2.2 is rather vague. For example, I assume much of the old relative recordings before it became digital were photographic recordings, but that is never mentioned (p9, l.261-262). Was the CJ6 theodolite really used to measure horizontal intensity, not inclination? There are a lot of abbreviations/model names (such as CZM, DTZ, GSM-19T). Can some more information such as manufacturer be given? In particular if the manufacturer still exists, this should be done. To my knowledge, the two mentioned GSM instruments are made by GEM Systems (https://www.gemsys.ca/) . I don't know about many of the others, but if possible, such information (references or websites) should be given. Is the instrumentation perhaps documented in the observatory yearbooks? Then this should also be mentioned with references to the yearbooks.

**Response:** Thank you for your valuable comments and suggestions on the description of the instruments in Section 2.2.
We have newly added a table in section 2.1 (Table 2 in the revised version of the manuscript), which exhaustively lists all the instrument data adopted by the eight geomagnetic observatories. This information covers the instrument models, types, manufacturing countries and manufacturers, measurement parameters, the year of first deployment at various Chinese stations, recording methods, as well as which stations have previously used these specific instruments.

**Photographic Recordings**: You are correct in your assumption that much of the old relative recordings before the digital era were indeed photographic recordings. This was an oversight on our part not to mention it explicitly. We have revised the text to include this information (Table 2).

**CJ6 Theodolite**: Regarding the CJ6 theodolite, we have consulted relevant materials and confirmed that it is indeed used for measuring horizontal intensity and declination.

**Comment:** f) Somewhat related to the previous point: Did all observatories become digital at the same time? As digital data should be available from that time on (probably even 1 min data, not just hourly values? – it would also be useful to mention somewhere which of the observatories are in INTERMAGNET now) this point in time should be clearly given for each observatory (perhaps it could also be included in an extended Table 1?)

**Response:** We have added explanations regarding whether the observatories joined INTERMAGNET in Table 1, and included the timeline of station digitization in Table 2.

**Comment:** g) Somewhat related to point e): The presentation of the data is the central point of the manuscript. Section 2.3 deserves some more details, so that the reader understands which parts of the data have been compiled in which way: On p10, in line 284 annual reports are mentioned and later for the digitization process "digital images" of the data are mentioned. I assume that the continuous recordings before digital instruments were installed were recorded photographically. It should be stated more clearly that data were digitized from published numbers, not directly from the original recordings. When scanning these pages, did you try to use characters (number) recognition, or did you indeed just scan images? If I understand the sentence in line 289/290 correctly, character recognition did not work and all data were digitized manually?

**Response:** You are absolutely right. The continuous recordings before digital instruments were installed were recorded photographically. The data were digitized from published numerical values rather than directly from the original photographic recordings. Regarding the digitization process mentioned in lines 289/290, we attempted to utilize Optical Character Recognition (OCR) technology to extract numerical data from scanned images. However, the quality of the original documents posed significant challenges to this process, resulting in unsatisfactory recognition outcomes. Consequently, we decided to fully adopt manual data entry to ensure data accuracy.

We have explicated these in lines 296-303 in the section 2.3 of the revised version of the manuscript.

**Comment:** h) Regarding the detection and estimation of jumps, section 3: have all jumps related to observatory re-location been estimated in this way (including the time of the jump, which is known in these cases)? Or have there been comparative measurements at some of the sites so that jump values had been documented?

**Response:** We estimated all jumps related to the observatory's relocation in this way, which is clearly visible in the Figure 2 of the original manuscript. Corresponding supplementary explanations have been added in lines 296-303 in Section 3 of the revised version of the manuscript.

**Comment:** i) p16, l. 417: I don't understand how increasing interference in the observatory environment would cause a long-term trend. Wouldn't one rather expect stronger scatter, rather than a continuous systematic drift in that case? But indeed I also don't have a suggestion for an interpretation of the drift, as an instrument drift would normally be captured by the absolute measurements. Might there be some slow change of the absolute pillar?

**Response:** Continuous infrastructure construction is being carried out around the BJI observatories, with the minimum distance between the villa area and the geomagnetic absolute observation room being less than 100 meters. I speculate this is a significant contributing factor.

**Comment:** j) p17, discussion of the differences: I think one of the main factors of short-term variations in the differences are remaining contributions from external magnetic fields in the data. If you only do a data selection for night times, but not additionally for geomagnetic activity, the data will still contain influences from geomagnetic storms and I would expect to see variations in the differences with the solar cycle. This seems included in your second reason, but I think it is one of the main reasons and should be stated more clearly. For the other aspect, this would be that the IGRF doesn't describe the secular variation accurately, and the five-year piecewise linear secular variation is not the best representation of the continuous secular variation. This could also be mentioned in addition to general IGRF quality considerations. I don't really understand what is meant by "participation of observation data" in l. 438/439.

**Response:** We agree with your comments on the discussion of differences section on page 17. In lines 462-468 of the revised version of the manuscript, we have rephrased to ensure clarity and comprehensibility.

**Comment:** k) p22, lines 541 and 543: From the first sentence I understand that the 1949 event was detected elsewhere in the world, so why would it only be a national or even more isolated regional event?

**Response:** Our intention is to illustrate that, given China had only the SSH observatory at that time, we could not determine whether this jerk event was prevalent across the entire country or was a phenomenon unique to that particular station. There were inaccuracies in the original manuscript, which we have now rephrased (in lines 569-571).

**Comment:** l) In Table 3, it would be helpful to distinguish between "no data available" and "data available, but no jerk detected" instead of having blank spaces for both. I suggest to use to different symbols (or blanks for no data, and a "-" symbol for data, but no jerk) or something like that to distinguish the two cases.

**Response:** In Table 3 (Now Table 4 in revised manuscript), we have updated the presentation to distinguish between "no data available" and "data available, but no jerk detected." We have used "/" to represent no data and a "-" symbol to indicate data available with no jerk detected.

The quality/resolution of all figures is rather low in the downloaded pdf. It should be checked if this is just a problem of the review version of the manuscript and improved if necessary.

**Response:** Thank you for your reminder. We will submit figures in *.eps, * format.

**Data set:**

**Comment:** 1. It would be more convenient for the user to have the three components, i.e. full magnetic field vector information, per time stamp for one observatory in one file. For easy compatibility with other observatories' data it would be best to follow the internationally adopted IAGA2002 format suggested by the International Association of Geomagnetism and Aeronomy (see publication by IAGA Working Group V-DAT at https://www.ncei.noaa.gov/services/world-data-system/v-dat-working-group/iaga-2002-data-exchange-format ).

**Response:** We have changed the dataset to the IAGA format and republished it on Zenodo (https://doi.org/10.5281/zenodo.15481895).

**Comment:** 2. The content of the files doesn't seem to be described. This should be done even if it is obvious that the first columns are date and time, and the last column is the magnetic component, but it is not clear to me what the number in the column in between is, it seems like just a counter that starts with 0 and I don't understand what it is useful for.

**Response:** We have changed the dataset to the IAGA format and republished it on Zenodo (https://doi.org/10.5281/zenodo.15481895).

**Comment:** 3. And as already stated above, I would strongly suggest to also submit the dataset to the World Data Center for Geomagnetism in Edinburgh, where hourly mean values from many geomagnetic observatories around the world is held.

**Response:** Thank you very much for your suggestion. We have already sent the data via email to the World Data Centre for Geomagnetism located in Edinburgh.

**Comment:** 4. Minor details of wording etc. in the manuscript:

-p1, l. 17: delete the word "rules"
-p2, l.70: delete the word "observation" before "observatories"
-p4, l.120: "observations" (plural)
-p5, l.158: Mention in the first sentence of the paragraph when this problem occurred
-p6, l.173: "observations" instead of "observational"
-p7,l.206: "resumed" instead of "resuming"
-p7, l.208: "It is located" instead of "The is located"
- the titles of sections 2.1.7 and 2.1.8 have wrong observatory names
-p8, l.241: I assume the sentence should say "observatories' yearbooks", or "in each

observatory's yearbook"

-p12, l.350: I suggest to use "contributions" instead of "components" here

-p13, l.378: "each observatory" (singular)

-p14 give a reference for the IGRF (e.g., the latest publication which currently is for IGRF 13th generation)

-p14, l.389: The IGRF describes the core or main field as detectable at Earth's surface, not just its long-wavelength components (shorter wavelengths do not play a role for practical purposes or any of the applications of the IGRF, because they are not measurable either due to their distance from the source or due to masking by the lithospheric field)

- p22, l. 538: "may have occurred"

**Response:** These detailed issues have been revised in accordance with your suggestions.

**Comment:** 5. Fig 6: The figure caption should contain more details of what is shown in the upper and lower panels, in particular the lower panel histograms should be briefly explained.

**Response:** Thank you for your valuable feedback. We have revised the caption of Figure 6 (Now Figure 7 in revised manuscript) to provide more detailed descriptions of both the upper and lower panels.

**Comment:** 6 Fig 7: Similarly, I suggest to include the information that now is given in each panel in the legend only once in the figure caption. I don't see the pink arrows mentioned in the legends.

**Response:** We have revised Figure 7 (Now Figure 8 in revised manuscript) according to your suggestions.